# ROS-Scavengers, Osmoprotectants and Violaxanthin De-Epoxidation in Salt-Stressed *Arabidopsis thaliana* with Different Tocopherol Composition

**DOI:** 10.3390/ijms222111370

**Published:** 2021-10-21

**Authors:** Ewa Surówka, Dariusz Latowski, Michał Dziurka, Magdalena Rys, Anna Maksymowicz, Iwona Żur, Monika Olchawa-Pajor, Christine Desel, Monika Krzewska, Zbigniew Miszalski

**Affiliations:** 1The Franciszek Górski Institute of Plant Physiology of the Polish Academy of Sciences, ul. Niezapominajek 21, 30-239 Kraków, Poland; m.dziurka@ifr-pan.edu.pl (M.D.); m.rys@ifr-pan.edu.pl (M.R.); amaksymowicz@ifr-pan.edu.pl (A.M.); i.zur@ifr-pan.edu.pl (I.Ż); m.krzewska@ifr-pan.edu.pl (M.K.); 2Faculty of Biochemistry, Biophysics and Biotechnology of the Jagiellonian University, ul. Gronostajowa 7, 30-387 Kraków, Poland; 3Department of Environmental Protection, Faculty of Mathematics and Natural Sciences, University of Applied Sciences in Tarnow, Mickiewicza 8, 33-100 Tarnów, Poland; m_olchawa@pwsztar.edu.pl; 4Botanical Institute of the Christian-Albrechts-Universität zu Kiel, Am Botanischen Garten 1-9, 24118 Kiel, Germany; cdesel@bot.uni-kiel.de; 5W. Szafer Institute of Botany, Polish Academy of Sciences, ul. Lubicz 46, 31-512 Kraków, Poland; z.miszalski@botany.pl

**Keywords:** antioxidants, carbohydrates, carotenoids, xanthophyll cycle, osmoprotectants, oxidative stress, ROS-scavengers, salt stress, α-/γ-tocopherols

## Abstract

To determine the role of α- and γ-tocopherol (TC), this study compared the response to salt stress (200 mM NaCl) in wild type (WT) *Arabidopsis thaliana* (L.) Heynh. And its two mutants: (1) totally TC-deficient *vte1*; (2) *vte4* accumulating γ-TC instead of α-TC; and (3) *tmt* transgenic line overaccumulating α-TC. Raman spectra revealed that salt-exposed α-TC accumulating plants were more flexible in regulating chlorophyll, carotenoid and polysaccharide levels than TC deficient mutants, while the plants overaccumulating γ-TC had the lowest levels of these biocompounds. Tocopherol composition and NaCl concentration affected xanthophyll cycle by changing the rate of violaxanthin de-epoxidation and zeaxanthin formation. NaCl treated plants with altered TC composition accumulated less oligosaccharides than WT plants. α-TC deficient plants increased their oligosaccharide levels and reduced maltose amount, while excessive accumulation of α-TC corresponded with enhanced amounts of maltose. Salt-stressed TC-deficient mutants and *tmt* transgenic line exhibited greater proline levels than WT plants, lower chlorogenic acid levels, and lower activity of catalase and peroxidases. α-TC accumulating plants produced more methylated proline- and glycine- betaines, and showed greater activity of superoxide dismutase than γ-TC deficient plants. Under salt stress, α-TC demonstrated a stronger regulatory effect on carbon- and nitrogen-related metabolites reorganization and modulation of antioxidant patterns than γ-TC. This suggested different links of α- and γ-TCs with various metabolic pathways via various functions and metabolic loops.

## 1. Introduction

Due to human activity and global climate changes, the area of heavily salinized (>2000 ppm) lands is on the increase. By 2050, it is expected to reach about 14% of global lands [1,2].

Salt (NaCl) stress induces ionic imbalance, and osmotic, and oxidative disturbances that affect many physiological processes in several subcellular compartments such as mitochondria [3] and chloroplasts [4,5,6]. Changes taking place in these organelles, and their impact on other subcellular compartments (e.g., peroxisomes [6,7]) initiate defense signaling pathways and regulate key metabolic processes.

Chloroplasts, through the shikimate/phenylpropanoid pathway, take part in the biosynthesis of chorismate, a precursor of aromatic amino acids (e.g., L-phenylalanine and L-tyrosine), and further intermediates, such as homogentisic acid (HGA, polar precursor of tocopherols), as well as a number of phenolic compounds (Figure 1). Through the 2-C-Methyl-D-erythritol 4-phosphate (MEP) pathway, chloroplasts are the sites of biosynthesis of geranylgeranyl pyrophosphate (GGPP), an intermediate in the biosynthesis of (poly)isoprenoids, such as e.g., carotenoids (CARs), chlorophylls, plastoquinol-9 or lipophilic polyprenyl—a precursor of tocopherols (TC) [8,9,10,11,12,13,14]. All of these metabolites are crucial for the molecular and physiological regulation of plant cell functioning, especially under stress conditions.

Tocopherols (methylated phenols, vitamin E) and carotenoids are two the most abundant groups of non-enzymatic lipophilic antioxidants in plastids. They both affect the physical and biochemical properties of lipid membranes [9,10], and their accumulation reflects salt stress tolerance [15,16,17,18]. TCs and CARs: (i) perform synergistic protective functions (e.g., remove reactive oxygen species, lipid soluble by-products of oxidative stress); (ii) help to maintain the balance between various metabolites/biochemical pathways; and (iii) share the same precursors/intermediates (e.g., GGPP) [9,10,11,13].

TCs can occur into four forms (α-, β- γ- or δ-) that differ in the position of methyl groups in the chromanol ring. TCs participate in removing reactive oxygen species (ROS) (e.g., ^1^O_2_, O_2_^•−^), protecting chloroplast membranes from photooxidation, regulating metabolite biosynthesis and gene expression, and are also components of the information-rich redox buffer system [9,13,19,20,21,22].
ijms-22-11370-sch001_Scheme 1Scheme 1Simplified scheme of metabolic pathways with marked metabolites the accumulation of which was affected by tocopherol (α-- and and γ-) levels. The metabolites marked in red are described in this manuscript, whereas those marked in green were examined in our previous publication [21].
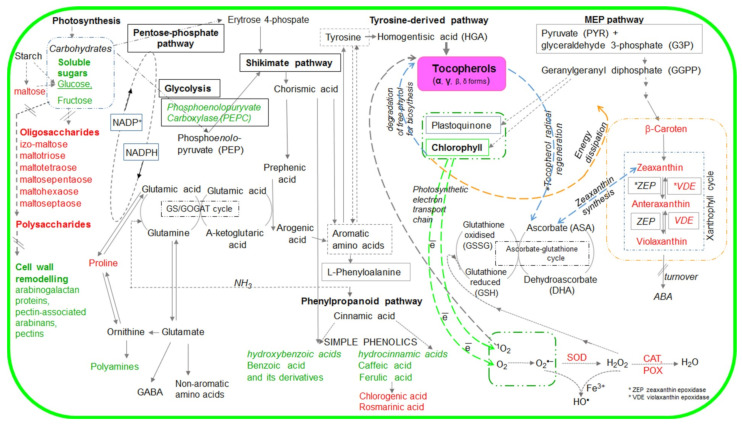


CARs are classified into two main groups: unoxygenated carotenes and oxygenated derivatives – polar xanthophylls [e.g., violaxanthin (Vx), antheraxanthin (Ax), zeaxanthin (Zx)]. The majority of CARs are located in functional pigment-binding proteins embedded in the thylakoid membranes. Among them, carotenes (mainly β-carotene) are bound to the photosystem reaction centers, while xanthophylls are the most abundant in the light-harvesting complexes. CARs play an indispensable role in energy transfer or dissipation of excess excitation energy, photoprotection by efficient quenching of chlorophyll triplate states, and scavenging ROS or other free radicals [10,11,23]. One of the most common CAR-controlled photoprotective process is xanthophyll cycle, in which Vx is de-epoxidated to Zx, via Ax by Vx de-epoxidase (VDE) under light stress [24,25].

Phenolics belonging to hydroxybenzoic and hydroxycinnamic acids are also involved in ROS regulation. They can function as signaling molecules, precursors of other stress-related, or structural compounds [8,12,14,26].

Additional important compounds determining plant salt stress tolerance include carbohydrates, sugar alcohols [27,28,29], and hydrophobic compatible solutes (osmoprotectants) including proline (Pro) [30,31], proline-betaine (PB) [8], and glycine betaine (GB) [31,32,33]. They are involved in the regulation of cellular water potential, (re)allocation of carbon and nitrogen, modulation of ROS and other free radicals, expression of stress response genes, and (de)activation of alternative detoxification pathways.

Primary and secondary metabolites interact with antioxidant enzymes, described as stress tolerance markers, which include superoxide dismutases (SODs, EC 1.14.1.1.) that catalyzes O_2_^•−^ disproportionation, and catalase (CAT, EC 1.11.1.6), which modulates levels of H_2_O_2_. These two antioxidant enzymes cooperate also with peroxidases catalyzing both ROS generation and scavenging [6,7,34].

Although the involvement of TCs in salt stress response was reported in several studies [16,21,35,36,37], more research is necessary to elucidate the metabolic loops, the links of TCs with different metabolites, and metabolic pathways controlling stress response. Changes in TC content were shown to affect accumulation of small molecular antioxidants [38] and carbon metabolites (e.g., carbohydrates, amino acids) [21,39] that are tightly linked to sulfur and nitrogen metabolism and crucial to plant stress tolerance [40,41]. Our previous study in *Arabidopsis thaliana* with altered TC composition exposed to salt stress showed a slight stimulation of the maximum operating efficiency accompanied by strongly altered cellular osmolarity [21]. Therefore, in this study we investigated the influence of TC composition on selected C-, and N-containing compounds with antioxidant and osmoprotectant properties that can be linked with TC based on their function or biosynthesis pathway (Figure 1). We also determined the effect of TC composition and salt stress on CAR profile and Vx de-epoxidation in the photoprotective reactions of the xanthophyll cycle. To distinguish between the role of α- and γ-TCs, we compared the response to salt stress under low light in the wild type (WT) *Arabidopsis thaliana* (L.) Heynh., its two mutants: (1) *vte1* totally TCs-deficient, (2) *vte4* accumulating γ-TC instead of α-TC, and (3) *tmt* transgenic line overaccumulating α-TC.

## 2. Results

### 2.1. Leaf Metabolites with Antioxidant and/or Osmotic Properties Detected by FT-Raman Spectroscopy

FT-Raman spectra obtained for the leaves of different Arabidopsis genotypes growing in control and salt stress conditions revealed several bands (Figure 1) denoting the presence of various chemical compounds in the plant tissues. The spectral pattern is characteristic of CARs with all-trans configuration of the conjugated C=C chain. The assignments of the three prominent Raman bands are well established: the 1005 cm^−1^ band is attributed to the C–CH_3_ rocking mode (CH_3_ groups attached to the polyene chain and coupled with C–C bonds), the 1156 cm^−1^ band to the C–C stretching mode/vibration (coupled with C–H in-plane bending), and the most intense band at 1525 cm^−1^ to the C=C stretching mode of the conjugated chain in CARs.

Changes in the amount of carotenoid compounds visible in the Raman spectra (bands at 1005 cm^−1^, 1156 cm^−1^, and 1525 cm^−1^) varied and depended on the type of mutants (*vte1*, *vte4*) or transgenic line (*tmt*) in both control and salt stress condition. The highest intensity band at 1525 cm^−1^, medium intensity band at 1156 cm^−1^, and the band at 1005 cm^−1^ showed in control *vte1* and *tmt* plants were higher, while in *vte4* mutants they were lower than in WT plants. The Raman spectra demonstrated that in control conditions the number of CARs in *vte1* and *tmt* plants was higher, and in *vte4* mutants lower, than in WT plants (Figure 1).

NaCl treatment changed the Raman spectral pattern characteristic of CARs. All plants with altered TC composition showed lowered band intensity, mainly at 1525 and 1156 cm^−1^, as compared with WT plants (Figure 1). Interestingly, a stronger decline was observed in *vte1* mutants and *tmt* transgenic line than in *vte4* mutants when compared with the controls. The bands at 1525 and 1156 cm^−1^ were almost unchanged in WT plants. Contrary to that, the lowest intensity band at 1005 cm^−1^ slightly declined in *tmt* transgenic line, strongly declined in *vte1* mutants, and remained fairly unchanged in *vte4* mutants and WT plants when compared to the control plants. Under salt stress, the decline in the amount of carotenoid compounds in *tmt* (~60%) and *vte1* (over 40%) plants was stronger than in *vte4* mutants. In contrast, in WT plants, CAR levels were comparable throughout the study under control and salt-stress conditions.

Under control conditions for all the tested plants, a very low intensity of the bands (at 1602, 1328, and 1270 cm^−1^) representing chlorophylls was observed. This is due to lower scattering of radiation by chlorophylls compared to carotenoids. The intensity of chlorophylls bands was slightly higher in *vte1* and *tmt* plants than in *vte4* and WT plants. NaCl treatment increased the intensity of these bands in WT plants, induced nearly no changes in mutants with α-TC deficiency, and decreased their intensity in the plants overaccumulating α-TC (Figure 1).

The bands visible at 1444 and 1304 cm^−1^ denote the deformation vibrations of CH, CH_2_, and CH_3_ groups and C-C stretching vibrations of aliphatic hydrocarbons, respectively. Control *vte1* and *tmt* plants showed slightly higher band intensity at 1444 and 1304 cm^−1^ than *vte4* and WT plants under the same conditions. Salt stress increased the intensity of 1444 and 1304 cm^−1^ in WT plants, and decreased it in the plants with altered TC composition. This decline was the strongest in *tmt* plants (Figure 1).

The bands (at 1188 and 851 cm^−1^) associated with polysaccharides are also visible in the Raman spectra. In control conditions, the intensity of these bands was higher in *vte1* and *tmt* mutants than in *vte4* and WT plants. NaCl treatment enhanced their intensity in WT plants, and reduced it in the plants with altered TC composition. This reduction was the most prominent in *tmt* plants (Figure 1).

### 2.2. Chemometrics–Cluster Analysis

Cluster analysis is a method of classifying tested objects into groups (clusters) so that the resulting clusters contain objects that are as similar as possible. We used the cluster analysis to find meaningful and systematic differences among the measured FT-Raman spectra, which show specific groups of the chemical compounds present in the plant tissues. The following dendrograms show the cluster analysis separately for two mutants (*vte1*, *vte4*), transgenic line (*tmt*), and WT plants grouping the control and salt-stressed plants (Figure 2).

Distinct discrimination was achieved for the two groups: control and NaCl stressed plants for the entire measuring range for each genotype (with or without α-TC and WT). It was demonstrated that the leaves of *vte1*, *vte4*, *tmt,* and WT plants differed significantly in their content and composition of carotenoids, chlorophylls, or polysaccharides and that these differences depended on salt stress conditions (Figure 2).

### 2.3. Vx De-Epaoxidation

Under control conditions, only *vte4* plants had higher initial rates of Vx de-epoxidation and Zx formation when compared with WT, while the values for *vte1* and *tmt* plants were comparable with those of WT plants (Figure 3A,B). Initial rates of Ax formation were similar in all tested plants (Figure 3C).

Control *vte4* plants were characterized by a greater drop in Vx (Figure 4A), and greater (8 percentage points (pp)) total Zx production than control WT plants (Figure 4B). No differences between maximal and final Ax levels were found in all tested groups of the control plants (Figure 4C).

Under salt stress, the initial rates of Vx de-epoxidation and Zx formation increased in WT, *vte1* and *tmt* plants but decreased in *vte4* mutants (Figure 3A,B). Significant changes were determined for total Vx loss, calculated as a difference in Vx level at the beginning and end of de-epoxidation, and for the final level of Vx and total Zx production after de-epoxidation between salt treated and control WT plants (Figure 4A,B). In control WT plants after de-epoxidation the total level of Vx dropped by about 50.20 ± 2.36 percentage points (pp) and that of Zx rose by about 45.87 ± 3.44 pp. In the salt-treated WT plants, the total reduction in Vx and rise in Zx were greater and reached 60.67 ± 2.13 pp and 56.12 ± 3.40 pp, respectively. The total decline and final level of Vx after de-epoxidation was similar in all tested experimental genotypes irrespective of the treatment (Figure 4A). No significant changes were also observed in the total decline and final level of Vx after de-epoxidation between the mutants (*vte1*, *vte4*), the transgenic *tmt* line and WT plants following exposure to salt stress. Under salt stress the only significant differences were spotted between WT plants and *tmt* transgenic line regarding a decline in total Zx production after de-epoxidation (Figure 4B). The highest initial rate of Ax formation among all tested plants was detected for salt-stressed *vte1* mutants (Figure 3C). Salt treated *vte1* plants showed also by about 2 pp higher maximal Ax accumulation than control *vte1* plants, whereas the differences between maximal Ax accumulation and the initial rate of Ax formation in the remaining variants were not significant, irrespective of NaCl presence. Salt-stressed *vte1* mutants showed also the highest differences between maximal and final level of Ax during de-epoxidation (Figure 4C).

### 2.4. Phenolic Acids

In control conditions the levels of chlorogenic acid (CGA) were comparable in WT, *vte1* and *vte4* plants and significantly lower in *tmt* plants (Figure 5A). Rosmarinic acid (RA) was higher in all plants with altered TC composition than in WT plants (Figure 5B).

Under salt stress conditions, the plants with altered TC composition produced less CGA than WT plants (Figure 5A). CGA content in salt-treated WT, *vte1* and *tmt* plants remained unchanged in comparison with their controls, while in *vte4* mutants its drastic drop was detected. NaCl treatment did not affect RA amount in WT plants but lowered its accumulation in plants with altered TC composition to the level comparable to WT plants (Figure 5B).

### 2.5. Carbohydrates and Polyols

Both under control and salt stress conditions, oligosaccharide content in plants with altered TC composition was lower than in WT plants. When comparing control plants, the lowest oligosaccharide amount was detected in *vte4* mutants (Figure 6A), and maltose content was significantly lower only in *tmt* plants (Figure 6B).

Salt treatment increased the content of oligosaccharides only in α-TC deficient mutants (*vte1*, *vte4*), and their levels remained unaffected in α-TC accumulating *tmt* plants when compared with the controls (Figure 6A). NaCl stress did not affect maltose level in WT plants but reduced its amount in α-TC deficient plants and enhanced it in *tmt* transgenic line (Figure 6B). Among salt-treated plants, *vte4* mutants showed the lowest, while *tmt* plants the highest maltose content.

Inositol content in control *vte4* and *tmt* plants was lower than in WT genotype (Figure 6C). Salt stress had no significant effect on its content in all tested plants, except for *vte4* mutants where the amount of this compound measured after NaCl treatment remained lower in comparison with WT plants.

### 2.6. Proline and Betaines

Proline (Pro) amount in all control plants was comparable (Figure 6D). Lower accumulation of proline betaine (PB) was detected in α-TC deficient plants when compared with α-TC accumulating plants and WT plants (Figure 6E). Glycine betaine (GB) amount in WT, *vte4* and *tmt* plants was comparable, and significantly higher in *vte1* mutants (Figure 6F).

NaCl treatment strongly boosted Pro content in all tested plants, with a higher amount being recorded in plants with altered TC composition than in WT plants (Figure 6D). Under salt stress, PB content declined in *vte4* plants, while the remaining genotypes were unaffected by NaCl (Figure 6E). α-TC deficient plants (*vte1*, *vte4*) showed a tendency to accumulate more Pro and less PB than *tmt* and WT plants. NaCl presence also strongly declined GB content in all tested plants when compared with their controls (Figure 6F), and this drop was greater in α-TC deficient plants (~10-fold decrease for *vte1*, ~12-fold for *vte4*) than in α-TC accumulating plants (~3-fold for WT, ~2-fold for *tmt*). Among all tested plants, the strongest NaCl-dependent decline and the lowest amount of PB and GB was observed in γ-TC overaccumulating plants (*vte4*).

### 2.7. Activity of Antioxidant Enzymes

In control conditions, total SOD activity in plants with altered TC composition was higher than in WT plants (Figure 7A).

The most pronounced activity among SOD isoforms was found for Fe-SOD and was similar in all tested plants (Figure 7B). The activity of Mn-SOD and CuZn-SOD was lower only in plants without α-TC. Also, the plants with altered TC composition had lower CAT activity than WT plants, with the lowest activity of this enzyme detected in *vte1* mutants (Figure 7C). Non-specific POX activity in plants with altered TC composition was comparable to WT plants (Figure 7D), and the plants overaccumulating α- or γ-TC (*tmt*, *vte4*) had lower total POX activity than *vte1* mutants.

NaCl treatment enhanced total SOD activity in all tested plants. The increase was the highest in WT plants, and higher in α-TC accumulating transgenic *tmt* line than in α-TC deficient plants (Figure 7A). NaCl treatment increased also the activity of Mn-SOD and two CuZn-SOD isoforms (CuZn-SODI, CuZn-SODII), more strongly in α-TC accumulating plants (WT, *tmt*), than in α-TC deficient plants (*vte1*, *vte4*; Figure 7B). Plants overaccumulating γ-TC showed the lowest intensity of Fe-SOD. Moreover, NaCl treatment resulted in a decline in CAT activity and an increase in non-specific POX activity in all studied plants as compared with their controls and WT plants (Figure 7C,D). NaCl treated plants can be ranked according to their CAT activity in the following order: WT> *vte1* = *vte4* = *tmt*, and according to their POX activity as: WT > *vte1* > *vte4* > *tmt*.

## 3. Discussion

Although a salinity of 100 *–* 200 mM has been proved toxic for most glycophytes [5,42], salt-induced lethality was not observed in plants growing under salt stress and low light intensity [21,43]. Our studies proved that the changed α- and γ-TC content is not a critical prerequisite for the survival of *A. thaliana* plants growing under salt stress (200 mM NaCl) and light intensity not exceeding 120 μmol photons m^−2^ s^−1^. *A. thaliana* plants with altered TC composition were capable of adapting to NaCl stress through modulation of their defense mechanisms that include not only TCs as quenchers of singlet oxygen formed at PSII [16,21,35,36,37,44].

### 3.1. Tocopherol Composition Modulates the Level of Photosynthetic Pigments (Chlorophylls, Carotenoids) and Intensity of Vx De-Epoxidation

The Raman spectra showing changes in the number of chlorophylls and carotenoids that depended on TC composition in control, and salt stressed plants (Figure 1 and Figure 2) revealed that mutual (α and γ) TC balance is important for the adjustment of the photosynthetically active pigments under both control and salt stress conditions. The changes in the level of photosynthetic pigments (chlorophylls, CARs) in *A. thaliana* plants with different TC composition subjected to salt stress (Figure 1) were accompanied by a slight stimulation of the maximum quantum yield of PSII [21]. We therefore suppose that a dynamic reorganization/recycling of metabolites related to TC enabled the plants to survive salt stress. The interdependence of TCs and chlorophylls is probably related to chlorophyll and protein degradation as well as *de novo* synthesis of TC precursors. These processes contribute to the increased demand for substrates used in TC synthesis during abiotic stress [22]. As showed by Ischebeck et al. [45] and Dörmann [46], free phytol from chlorophyll breakdown under stress might be directly used for TC biosynthesis. Moreover, α- and γ-TC dependent changes in chlorophyll and CAR accumulation (Figure 1) can be linked with various α- and γ-TC physical and chemical traits (e.g., stronger nucleophilic properties of γ-TC than α-TC), and their different membrane or cellular location [13,19]. We also found that under salt stress α-TC enabled more efficient modulation of signals related to photosynthetic pigment (chlorophylls, CARs) biosynthesis or accumulation than γ-TC. As the maximum quantum yield of PSII (Fv/Fm, stress indicator) in salt-stressed *A. thaliana* with altered TC composition was slightly stimulated [21], we speculated that TC dependent changes in CAR composition were related not only to the antioxidant mechanism of TCs [9,19,21] and CARs [24,47,48] protecting physical properties of the membranes, but probably also resulted from downregulation/alteration of CAR biosynthesis (as regulation of shikimate-phenylpropanoid and MEP pathways). The heterogeneity of plants with or without α-TC (Figure 2) is due to their chemical composition, especially the content of CARs, chlorophylls, polysaccharides, and compounds with CH, CH_2_, and CH_3_ groups and C-C or C=C bonds.

Oxygenated derivatives of CARs, such as Vx, Ax, and Zx are involved in the violaxanthin cycle, which represents an important element of plant cell stress defense strategy, including high light intensity and salinity. The interdependent processes of epoxidation and particularly de-epoxidation (removal of two epoxide groups of Vx in two steps by VDE, forming first a monoepoxide Ax, and finally a completely epoxide free Zx) are crucial for photoprotection of chloroplasts. The formation of Zx and Ax was found to correlate with dissipation of excess excitation energy (non-photochemical quenching; NPQ) of chlorophyll fluorescence [11,25]. The intensity of Vx de-epoxidation, in addition to light stress, can be enhanced by other synergistically interacting environmental stress factors, including salinity [11,15]. Our results confirmed that initial rates of Vx de-epoxidation and Zx formation were higher in salt-stressed than in control WT plants (Figure 3A,B), and indicated that the naturally balanced level of α-/γ-TCs (WT plants) played an important role in optimizing the rate of Vx de-epoxidation and Zx formation (Figure 4A,B). Altered TC composition, particularly the lack of α-TCs modified various steps of Vx de-epoxidation, and caused a temporary change in ROS levels even if high light intensity was the only stress factor. Stronger differences in various steps of Vx de-epoxidation in salt-stressed α-TC deficient plants than in α-TC accumulating plants (Figure 3 and Figure 4), indicated a more important role of α-TC than γ-TC in regulating Zx formation under high light and salinity (Figure 3A,B). In addition, the lowest CAR level and changes in Vx de-epoxidation in salt-stressed mutants overaccumulating γ-TC, may be related to the lack of α-TC and accumulation of intermediate metabolites formed by blocking α-TC from γ-TC synthesis pathway. Contrary to that, it seemed that double TC mutants (with knockout in tocopherol cyclase 1 gene catalyzing the penultimate step of TC synthesis [38]) presumably protected photosynthetic complexes from oxidative stress through tocotrienols [19], as tocopherol cyclase 1 is involved in the biosynthesis of both TCs and tocotrienols. Moreover, the lack of both TC forms, together with reduced level of CARs under salt stress (Figure 1), should facilitate de-epoxidation as TCs, tocotrienols, and CARs regulate the molecular dynamics and physical properties of the thylakoid membranes [10,19] in which Vx de-epoxidation occurs. Low amounts of TCs and CARs in salt-treated double TC mutants (Figure 1 and Figure 2) can result in higher mobility of the membrane, increased diffusion rate of Vx, and higher initial rate of de-epoxidation followed by the most intensive accumulation of Ax (Figure 3C and Figure 4C). In addition, our findings (Figure 3 and Figure 4) confirmed that salinity affected xanthophyll cycle functioning independently of changes in the genes involved in TC biosynthesis. α- and γ-TCs seemed to selectively optimize Vx and Ax de-epoxidation (Figure 1, Figure 2, Figure 3 and Figure 4), which might to a different degree affect the photoprotective and antioxidant functions of these molecules [11,23,24,47], their interactions with the components of antioxidant network governed by the ascorbate-glutathione cycle [13,49], and differentially modulate physical properties of lipid membranes [10,50]. Changes in Vx, Ax, and Zx profiles (Figure 3 and Figure 4) may have reflected TC dependent modulations of metabolic pathways, including cytosolic mevalonate and plastid MEP pathways [11,48,51], (Figure 1).

They could also indicate alterations in the biosynthesis of the precursors or derivatives of xanthophyls, including apocarotenoids (e.g., abscisic acid, ABA) [48] that can be involved in the regulation of TC biosynthesis in Arabidopsis exposed to abiotic stresses [52]. According to Ellouzi et al. [37], salt-stressed Arabidopsis plants overaccumulating γ-TC are not as efficient in modulating ABA accumulation as double TC mutants and WT plants. TC dependent reorganization of photoprotective pigment (carotenes, xanthophylls) accumulation in chloroplasts (Figure 1, Figure 2, Figure 3 and Figure 4) seems to reflect their functional and metabolic relationships.

### 3.2. Tocopherols Affected Energy Use and Dissipation

We found that both the phenolics such as cinnamic and ferulic acids [21] and their derivatives such as CGAs [53] and RA [26] were influenced by TC composition in control and salt stress conditions (Figure 5A,B). TC related modification of phenylpropanoid accumulation and probably their interactions with other biocompounds (including CGA or RA precursors, derivatives, intermediates) (Figure 1), might affect the dynamics of cellular antioxidant potential, adaptive signaling pathways, photochemical energy dissipation possibilities, and membrane properties [14]. TC dependent changes in CGA content (Figure 5A) might affect the mitochondrial tricarboxylic acid cycle (TCA) and amino acid metabolism [54,55].

### 3.3. Tocopherol Composition modulates the Pool of Primary and Secondary Metabolites under Salt Stress

Alterations in carbohydrate metabolism belong to salt stress response mechanisms [27,29,56]. Under salt stress, soluble sugar accumulation is influenced by TC composition [21,39] (Figure 1 and Figure 6). Based on the Raman spectra, we stated that salt stress-triggered accumulation of polysaccharides localized in the cell wall or epidermis also depended on α- and γ-TC content (Figure 1), as α-TC containing plants adjusted cell wall or epidermis polysaccharide level more flexibly than γ-TC deficient plants. These results corresponded with *α*-TC dependent expression of several pectin and AGP epitopes in the leaves of salt-treated Arabidopsis [21]. Contrary to that, the fact that soluble oligosaccharides (hexose equivalents) were accumulated only in salt-stressed α-TC deficient plants (Figure 6A) suggested a reduced demand for soluble sugars. This was probably a consequence of shoot growth limitation [57], as well as various roles of α- and γ-TC in controlling the accumulation of osmotically active sugars involved in the regulation of membrane or cell wall potential [21] or sugars being used for stress-dependent cell wall reorganization.

TC composition affected the level of maltose (Figure 6B), the major product of starch degradation in chloroplasts. Maltose is also a source of glucosyl residue that is converted to hexose phosphate in the cytosol [27,56,58]. We only saw increased maltose content in salt-stressed plants overaccumulating α-TC, and its reduction in α-TC deficient plants (Figure 6B). These findings corresponded to those of Abbassi et al. [35], who reported on TC dependent changes in starch levels in salt treated transgenic (*vte2*, *vte4*) *Nicotiana tabacum*, and described modulation of maltose-related processes. Elevated maltose content in the plastids of freeze stressed *A. thaliana* was proposed as a mechanism protecting the photosynthetic electron transport chain, proteins, membranes [27], and biosynthesis of the compounds involved in cell walls reorganization [28].

A simultaneous reduction in inositol, particularly in γ-TC overaccumulating plants (Figure 6C), revealed different roles of α- and γ-TC in inositol-related processes (i.e., ascorbic acid biosynthesis) and signaling intricately tied to lipids [59].

Our study showed that under salt stress conditions, TC composition affected accumulation of polyamines [21] and of Pro and methylated amino acids derivatives (PB, GB) (Figure 6D–F). These are biocompounds involved in the Glu (glutamate)-Pro-Arg-PAs-GABA (γ-aminobutyric acid) pathway localized, at least partly, in chloroplasts and are responsible for (re)allocating of assimilated C and N [60]. This is in agreement with the studies showing that TC composition affects total amino acid and Pro levels in transgenic *N. tabacum* [35] and the accumulation of some non-aromatic amino acids in the leaves of TC deficient transgenic tomato [39]. As the accumulation of Pro, one of the first osmoprotectants activated under environmental stresses [30,31,61] was more intense in the salt-treated plants with altered TC composition (particularly without either TC form) than in WT plants (Figure 6D). We suggested that Pro may partially compensate the changes in TC composition and may reflect elevated carbon flux from the carboxylate into the amino acid pool. Pro also serves as a C, N, and energy source for the cellular recovery processes or metabolic pathways in which also TCs can be involved [30,31,60]. Pro biosynthesis in Arabidopsis subjected to salinity is photosynthetically (-light)-dependent and is inhibited by terpenoid or isoprenoid groups (e.g., brassinosteroids [62]). The biosynthesis or accumulation of Pro are mediated by signaling pathways dependent and independent of ABA, linked with glucose oxidation and oxidative pentose phosphate pathway. Under salinity, Pro biosynthesis is raised in the plastids, although under normal conditions proper Pro concentration is maintained in the cytosol [8,31,60,61]. According to Signorelli [30], an increased rate of Pro biosynthesis or accumulation through NADP^+^ re-oxidation may prevent photosynthetic electron leakage and ROS generation. Also, through the generation of NADH/H^+^, FADH_2_ and ATP in the mitochondria Pro promotes cell survival under stress conditions. Pro can affect the initial stages of phenylpropanoid and protein biosynthesis [8,63,64], controls the expression of salt stress responsive genes (e.g., PRE, ACTCAT) [60] and the activity of some antioxidants, affects cellular redox buffering [64] and modulates cell wall architecture [63]. On the other hand, Pro accumulation may lead to protein denaturation [65] and may initiate programmed cell death [66].

Our study indicated that α-TC levels under salinity also affected the accumulation of methylated amino acid derivatives (PB or GB) and PB or GB related processes (Figure 6E,F). For instance, GB takes part in the regulation of ROS level and Na^+^/K^+^ homeostasis [31,32,33], depending on TC composition [37]. It seems that under salt stress, the accumulation pattern of Pro and methylated amino acid derivatives (GB, PB) was linked with the level of chromanol ring methylation in α-TC and γ-TC. The methylation reaction was shown as being involved in the regulation of gene expression, and/or biosynthesis of stress-response biomolecules, including TCs [9], Pro, betaines (e.g., GB), phenolic compounds, chlorophylls, or plastoquinones [31,33,67]. The association between salt stress tolerance and methylation was reported (e.g., by Kumar et al. [68], who found salt-induced tissue-specific cytosine methylation in *Triticum aestivum*, and by Karan et al. [69], who indicated salt-induced variation in DNA methylation pattern and gene expression in rice). In addition, nitrogen metabolism rearrangement was associated with the dissipation of the excess light energy through xanthophyll cycle [70], and the composition of the pool of xanthophyll cycle pigments [71], whose de-epoxidation under saline conditions depends on α-/γ-TC ratio (Figure 3 and Figure 4). We suggest that α- and γ-TC dependent carbon and nitrogen biocompounds accumulation revealed differences in the regulation of C/N-related metabolic pathways (compartmentalized between chloroplast, cytoplasm and mitochondria) and the energy status of the cell.

The ROS balance and ROS-associated redox signals, described as crucial for harmonious metabolism and the establishment of adaptive signaling pathways, are maintained in the cells mainly through the mutual cooperation of non-enzymatic (e.g., TCs, CARs) and enzymatic (SOD, CAT, POX) antioxidants [6,14,19,20,34,72]. In agreement with our previous studies on Arabidopsis seedlings [16], we showed that changes in α- and γ-TC content were differentially compensated by alterations in SOD, CAT (Figure 7A–C), and POX (Figure 7D) activity in both control and salt stress conditions. It was shown previously that the exposure of TC deficient Arabidopsis mutants to high light [20,72], and salinity Ellouzi et al. [37] boosted the oxidative stress as compared with WT plants with naturally balanced TC levels. The changes in SOD activity can be related to α- and γ-TC content through their influence on the substrate level for SOD isoforms in different cellular organelles, and thus the modulation of ROS-dependent signal induction under salt stress conditions. The inactivation of CuZn-SOD observed in α-TC deficient plants could be explained by intensive NaCl induced ROS generation. This suggests an important role of synergistic interaction of α-TC and SODs in cell protection against oxidative damage.

Total SOD activity pattern did not closely correspond with the patterns of CAT and POX activity (Figure 7A,C,D), which may suggest α- and γ-TC dependent differences in the level of substrates for CAT and POX generation and specific functions of these antioxidant enzymes. CAT (widely used as a peroxisomal marker) is crucial in removing photorespiratory H_2_O_2_ [7] and may also reflect other peroxisomal metabolic pathways linked with nitrogen metabolism, β-oxidation of fatty acids and biosynthesis of phytohormones (e.g., indolilo-3-acetic acid (IAA) and jasmonic acid (JA) [73]). Munne-Bosch et al. [74] showed TC dependent synthesis of JA, a precursor of which, 12-oxophytodienoicacid, is generated in chloroplasts and subsequently synthesized in peroxisomes via β-oxidation. TC dependent changes in POX activity (Figure 7D), the level of Pro (Figure 6D), carbohydrates and phenolics [21] (Figure 5 and Figure 6A) can be associated with TC related cell wall modification under salt stress [21]. Our results also suggested TC related involvement of CAT and POX in H_2_O_2_-dependent signaling pathway in the cellular processes compensating for altered TC balance and salinity response.

## 4. Materials and Methods

### 4.1. Plant Material

The seeds of *A. thaliana*: Columbia ecotype (Col-0) (1), homozygous mutant *vte1* (GABI_111D07) in the Col background with an insertion in the third intron of the open-reading frame (At4g32770) of the gene encoding tocopherol cyclase; deficient in tocopherol cyclase, totally devoid of TCs (2), the homozygous mutant *vte4* (SALK_036736) in the Col background with an insertion in the first intron of the open-reading frame (At1g64970) of the gene encoding γ-tocopherol methyltransferase; devoid of γ-tocopherol methyltransferase (γ-*TMT*, catalyses the conversion γ-TCs to α-TC) gene, accumulating γ- instead of α-TC (3), and transgenic γ-*TMT* plants overexpressing γ-*TMT* methyltransferase under the control of 35S CaMV promoter, thus overproducing α-TC (4), described by Desel et al. [75] and Fritscheet al. [9], were kindly provided by Prof. Karin Krupińska (Kiel University, Germany). The experimental plant line of *A. thaliana* used in our experiments has a well-documented origin Desel et al. [75] and was thoroughly characterized e.g., by Porfirova et al. [76], Bergmuller et al. [77], Shintani and DellaPenna [78], Collakova and DellaPenna [22], or Rosso et al. [79].

The seeds were germinated in the soil for three weeks and then each plant was transferred to a single pot. The six to seven-week-old plants were then divided into two groups. The first set of plants was subjected to rapid salinization by irrigation with 200 mM solution of NaCl for 10 days. The second group (i.e., the control plants) were irrigated with tap water. The plants grew under 100–120 μmol m^−2^ s^−1^ light intensity, at 18 °C, 12/12 h light/dark photoperiod, and 40–60% relative air humidity (RH). The plants were cultivated in a randomized design and rotated daily to minimize positional effects. Each group within the genotype consisted of at least 18 to 22 replicates (plants). For biochemical analyses the pooled samples were harvested from the representative rosette leaves of five to seven plants per genotype, growing under control or salt stress conditions. The samples were collected at the end of the light period (between 16.00 and 18.00) in three replicates. Three independent experiments were performed. Vx de-epoxidation was analyzed in 5 mm diameter leaf discs after 60 min dark incubation. The rest of the collected plant material was immediately frozen in liquid nitrogen (LN2) and stored at −80 °C until analysis.

The content of α- and γ-TC in the rosette leaves collected at the end of the experiment in both control and salt stress conditions was monitored according to a modified method of Surówka et al. [80] and presented in Table 1.

In WT plants, the levels of α-TC were comparable throughout the study under control and salt-stress treatment (Table 1). As expected, *vte4* mutant showed γ-TC but not α-TC accumulation in the rosette leaves, *vte1* mutant did not accumulate either α- or γ-TCs in the leaves, and *tmt* transgenic line showed α-TC, but not γ-TC accumulation in the rosette leaves in both control and salt stress conditions.

### 4.2. Fourier Transform Raman Spectroscopy Measurements and Chemometrics

The changes in chemical component profiles were assessed using a non-invasive technique—Fourier-transformation Raman spectroscopy (FT-Raman) [81,82]. The Raman spectra of *A. thaliana* leaves were recorded with a FT-Raman Spectrometer Nicolet NXR 9650 (Thermo Scientific, Walthman, MA, USA) equipped with a Nd:YAG 1064 nm laser and a germanium detector cooled with liquid nitrogen. The spectrometer was provided with a xy stage, a mirror objective and a prism slide for redirecting the laser beam. The spectra were collected in the range of 100 to 4000 cm^−1^ at 250 mW laser power with a 4 cm^−1^ resolution. Each spectrum included 128 scans. All spectra were registered by the Omnic/Thermo Scientific software. The leaves of *A. thaliana* were lyophilized. One leaf from each collected sample was used for the measurements. Five spectra were collected for the leaf, and then the baseline was corrected and averaged.

### 4.3. Violaxanthin De-Epoxidation

Leaf discs 5 mm in diameter were placed in the Petri dishes lined with wet filter paper. The discs were illuminated with 1700 μmol m^−2^ s^−1^ for 0; 5; 10; 20; 30 and 40 min. Then the material was frozen in LN2 and stored at −80 °C until analyzing by high performance liquid chromatography (HPLC, Agilent 1260 Infinity system, Waldbronn, Germany) as described by Latowski et al. [83].

### 4.4. Assays for ROS Scavengers and Osmoprotectants

#### 4.4.1. Analysis of Phenolic Compounds

Phenylpropanoid compounds, such as chlorogenic acid (CGA) or rosmarinic acid (RA) were estimated according to Hura et al. [84] and Surówka et al. [21]. Two sets of dynamically modified excitation (Ex) and emission (Em) wavelengths, for chlorogenic acid (CGA; Ex 325 and Em 424 mn) and rosmarinic acid (RA; Ex 330 and Em 410 nm), were used for the fluorimetric detection. For further technical details, please see Golebiowska-Pikania et al. [85].

#### 4.4.2. Carbohydrates and Sugar Alcohols

Maltose and oligosaccharides (i.e., iso-maltose, maltotriose, maltotetraose, maltosepentaose, maltohexaose, maltoseptaose) determined as hexose equivalents [nmol/mg FW]) and inositol were measured according to Hura et al. [84] and Surówka et al. [21].

#### 4.4.3. Proline and Betaine Estimation


*Proline*


Free proline was estimated by means of spectrophotometric method in 96-well-plate format (Synergy II, Biotek, Winoski, VT, USA), after derivatization with ninhydrine reagent as reported by Surówka et al. [80].


*Betaines*


Glycine betaine and stachydrine were estimated in lyophilized samples (≤0.02 g FW) as detailed by Wiszniewska et al. [86] with some modifications. [^2^H_4_]1-amino-1-cyclopropanecarboxylic acid (D-ACC) was added as an internal standard. UHPLC separation was performed by hydrophilic interaction liquid chromatography (HILIC) on an Agilent 1260 Infinity system (Agilent, Waldbronn, Germany). Separation was achieved on a Kinetex HILIC column (75 mm × 2.1 mm, 2.6 µm, Phenomenex, Torrance, CA, USA) at 35 °C. For detection, an Agilent Technologies 6410 triple quadruple mass spectrometer (MS/MS) equipped with electrospray ionization (ESI) in the positive ionization mode was used. Mass hunter software was used to control the UHPLC–MS/MS system and for data analysis. Pure GB and PB were used as external standards, a surrogate was an internal standard for GB and PB.

### 4.5. Protein Extraction and Determination of Antioxidant Enzyme Activity

Soluble proteins were isolated from the leaves (0.1 g FW) homogenized in a cooled mortar in 2.5 mL of an extraction buffer (3 mM MgSO_4_, 1 mM DTT, 3 mM EDTA, 100 mM Tricine, pH 8.0, TRIS). The supernatant obtained after centrifugation (20,000× *g*, 20 min) was used to determine the activity of the antioxidant enzymes.

Protein concentration was determined according to Bradford [87], with Bio-Rad Protein Assay (Bio-Rad, Hercules, CA, USA), and using bovine serum albumin (BSA) as a standard.

Total activity of SOD was determined according to Minami and Yoshikawa [88], with 50 mM TRIS-cacodylic buffer pH 8.2. The reaction mixture contained 0.1 mM EDTA, 1.4% (*v*/*v*) Triton X-100, 0.055 μM NBT, 16 μM pyrogallol and the plant extract. The reduced form of NBT was measured at 540 nm. A unit of enzyme activity [U] was defined as the enzyme activity that inhibits auto-oxidation of pyrogallol by 50% according to McCord and Fridovich [89].

CAT activity was evaluated according to Aebi [90], by monitoring the disappearance of H_2_O_2_ at 240 nm, in 50 mM phosphate buffer pH 7.0. The enzyme activity was determined in units [U] defined as 1 mmol of H_2_O_2_ degraded in 1 min per 1 mg of protein.

Non-specific peroxidase (POX) activity was measured according to Luck [91] by following the decomposition of p-phenylenediamine (pPD) H_2_O_2_-dependent for 2 min at 460 nm. The extinction coefficient of 1.545 × 103 M^−1^ cm^−1^ was used as described by Allgood and Perry [92]. Total peroxidase activity was described as nmol of pPD decomposed in 1 min per 1 mg of protein.

### 4.6. Analysis of SOD by Native PAGE

For determining SOD isoform activity, the fractions of soluble proteins were separated by native polyacrylamide gel electrophoresis (PAGE) on 12% gel at 4 °C, 180 V, in the Laemmli [93] buffer system without sodium dodecyl sulfate (SDS), as described previously by Miszalski et al. [94]. SOD bands were visualized using the staining procedure by Beauchamp and Fridovich [95]. SOD bands were analyzed densitometrically.

### 4.7. Statistical Analysis

Analysis of variance (ANOVA) and post-hoc Tukey’s multiple range test were performed to determine significant differences between *A. thaliana* genotypes and treatments at the significance set at *p* ≤ 0.05).

The similarities between the FT-Raman spectra were studied using a hierarchical cluster analysis (Statistica package 10). The spectra were baseline corrected. The cluster analysis was performed separately for WT, *vte1*, *vte4*, and *tmt* plants for the entire wavenumber range using Ward’s algorithm. The spectral distances for WT and *tmt* plants were calculated with the standard algorithm with previously unprocessed data. For *vte1* and *vte4* plants, the cluster analysis was carried out with the factorization algorithm using the first two factors followed by vector normalization.

## 5. Conclusions and Challenges

Under salt stress, α- and **γ**-TC content differentially influenced the induced compensatory mechanisms, including the components of both non-enzymatic (e.g., CARs, phenols) and enzymatic (SOD, CAT, PODs) antioxidant systems, as well as osmoprotective (Pro, GB, PB, carbohydrates) networks (Figure 1). The new cellular balance, achieved by α- and γ-TC dependent accumulation of metabolites of the shikimate-phenylpropanoid, MEP and the Glu (glutamate)-Pro-Arg-PAs-GABA (γ-aminobutyric acid) pathways involved reorganization of carbon and nitrogen metabolism and allowed for energy dissipation and ROS scavenging via alternative pathways.

Changes in the status of lipophilic α- and **γ**-TC influenced the level of CARs and the rate of de-epoxidation. It seems that not only the final level of accumulated xanthophylls but also the amount of transient xanthophylls and the interplay among these photosynthetic pigments at different stages of de-epoxidation depended on the α-/**γ**-TC ratio and might have important functions in plant signaling (including ABA and JA formation).

Under salt stress, α-TC appears to have a stronger regulatory effect on the pattern of accumulated biocompounds and de-epoxidation than γ-TC, which seems to inhibit some of the defense reactions. α-TC is responsible for key aspects of salt stress adaptation through modification of signals originating in the chloroplasts.

It seems that TCs function at the crossroad of ROS and methylation dependent processes, affecting carbon and nitrogen dependent metabolism. Altered content of differentially methylated α- and γ-TCs and modified TC biosynthetic pathways can affect cellular methylation processes linked probably with methionine cycle, but this statement requires further research.

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
