# Peer review of "ROS-Scavengers, Osmoprotectants and Violaxanthin De-Epoxidation in Salt-Stressed Arabidopsis thaliana with Different Tocopherol Composition"

_ijms, 2021, doi:10.3390/ijms222111370_

Round 1

Reviewer 1 Report

#Manuscript report

The manuscript entitled “Alternations in endogenous α- and γ-tocopherols changes parallel to ROS-scavengers, osmoprotectants patterns, and violaxanthin de-epoxidation under salt stress in Arabidopsis thaliana plants submitted by Surówka et al.. described the importance of tocopherol in salt stress tolerance. In the manuscript, they wrote results following the detailed discussion with a supporting conclusion. However, lots of required information is missing, such as molecular data; hence I cannot recommend this paper in IJMS as available in present form. Therefore, the Reviewer has several suggestions and comments are as follow:

Abstract: It is well-written =.

Introduction-

Line 87-modulating H2O2 and hydrogen sulphide level, which cooperate with peroxidases catalyzing both, ROS generation and scavenging [6,7,33].

I mean how catalase modulate hydrogen sulphide?

Results

Line 110-FT-Raman spectra obtained for leaves of different genotypes of Arabidopsis (Nonitalic) growing

Results-In the results section they did not follow the uniform pattern. It is very hard to get a clear message from the result part. I would like to suggest you please make it simple and interesting. Although, results are acceptable after following minor modifications.

Such as:

2.2. Chemometrics - Cluster analysis-Kindly elaborate it.

Line 180-Statistically significant changes in 181 the total decline, as well as final level of Vx after de-epoxidation and total 182 Zx production, were observed between salt treated and control WT plants 183 (Fig. 3 A, B). It's not a clear message to me. Kindly write in a proper way.

Line 187-The total decline and final level of Vx after de-epoxidation, were comparable with whom? among all group of plants subjected to salinity (Fig. 3A), while total 189 Zx production was lower in transgenic tmt line in comparison with WT plants which one? Salt treated WT or Controlled WT? (Fig. 3B).

Line 192-De-epoxidation rate was so high, that maximal level of Ax accumulation increased at c.c. 2 pp and was the largest, 194 among of tested plants (Fig. 3C). You just said higher or increased but in which genotype?

2.7. Antioxidant enzymes activities

In figure 6B-The gel pictures are a blur and not able to see the clear band. Additionally, Controls are missing from this image. Therefore, need to replace it with a new picture along with negative and positive controls.

Line 263-Low activity of MnSOD and CuZnSOD isoforms was also 264 detected.???? You have just landed the sentences. Kindly write it properly.

Line 279-Very similar is its rank in respect of POX, but showing an increase in activity (WT 280 vte1 vte4 tmt). What?? I did not get. Kindly rewrite it.

Discussion:

Line 284-Our studies also proved that A. thaliana plants are able to adapt to NaCl stress under light intensity How??

Line 289- Tocopherols composition modulate the level of photosynthetic pigments (chlorophylls, 290 carotenoids) and intensity of Vx-de-epoxidation

Where is the data of photosynthesis parameters of growing plants?

Line 291-Obtained Raman spectra revealed that mutual ( - and ) TCs balance play role 292 in photosynthetically active pigments (chlorophylls, CARs) adjustment under both con293 trol and salt stress conditions (Fig. 1A, B). According to Ischebeck et al. [44] and Dörmann 294 [45]. This statement is not supported by your results.

As you reported the role of these mutants in photosynthesis. What about the role of these in chloroplast morphology? Do you have electron microscopy pictures of chloroplast under control and salt stress?

Material methods:

Note-Kindly provides the transcript level of your transgenic lines under normal as well as under salt stress.

Author Response

Responses to Reviewer 1

Dear Sir/Madam,

We would like to thank you for taking the time to review our publication. We are very grateful for sharing your remarks with us. We found all of them extremely helpful. We also appreciate all minor remarks suggested with the aim to improve our paper. The manuscript was corrected by a professional linguist. Sentences corrected or added to the manuscript text as suggested by the Reviewer are highlighted in green. In the pdf-document we present our responses to your remarks.

Introduction

Line 87-modulating H2O2 and hydrogen sulphide level, which cooperate with peroxidases catalyzing both, ROS generation and scavenging [6,7,33].

I mean how catalase modulate hydrogen sulphide?

c

Catalase is regulator of reactive sulphur species (RSS) bioavailability. Under normoxic conditions, catalase can oxidize SO2, H2S, and H2Sn species. Conversely, catalase reduces sulfane sulfur species to H2S under hypoxic conditions (Kevil 2017). Catalase can act as a sulfide-sulfur oxido-reductase (Olson et al. 2017). Moreover, Corpas et al. (2019) provided evidence of the presence of H2S in Arabidopsis thaliana peroxisomes, which appears to regulate catalase activity and, consequently, the peroxisomal H2O2 metabolism. Authors suggested also that H2S can be regarded as a new regulatory molecule that may be involved in crosstalk between peroxisomes and other subcellular compartments, especially under nitro-oxidative stress conditions.

Christopher G. Kevil (2017) Catalase as a regulator of reactive sulfur metabolism; a new interpretation beyond hydrogen peroxide,Redox Biology,12, 528-529.

Corpas, FJ, Barroso, JB, González-Gordo, S, Muñoz-Vargas, MA, Palma, JM (2019) Hydrogen sulfide: A novel component in Arabidopsis peroxisomes which triggers catalase inhibition. J Integr Plant Biol 61: 871– 883.

Kenneth R. Olson, Yan Gao, Eric R. DeLeon, Maaz Arif, Faihaan Arif, Nitin Arora, Karl D. Straub (2017). Catalase as a sulfide-sulfur oxido-reductase: An ancient (and modern?) regulator of reactive sulfur species (RSS). Redox Biology, Volume 12, 325-339

However, we have decided to remove the fragment of sentence concerning the role of catalase in H2S metabolism, as it is not crucial in the discussion concerning tocopherol-dependent changes in carbon and nitrogen metabolism.

Results

Line 110-FT-Raman spectra obtained for leaves of different genotypes of Arabidopsis (Nonitalic) growing

We change the fonts in word “Arabidopsis”

In the results section they did not follow the uniform pattern. It is very hard to get a clear message from the result part. I would like to suggest you please make it simple and interesting. Although, results are acceptable after following minor modifications.

Such as:

We added information explaining the presence of the Raman bands.

FT-Raman spectra obtained for the leaves of different Arabidopsis genotypes growing in control and salt stress conditions revealed several bands (Fig. 1A) denoting the presence of various chemical compounds in the plant tissues.

We also modified the paragraph describing changes in carotenoids based on Raman spectra as follows:

Changes in the amount of carotenoid compounds visible in Raman spectra (bands at 1005cm−1, 1156cm−1 and 1525cm−1) are variable and depend on the type of mutants (vte1, vte4) or transgenic line (tmt) in both control and salt stress condition. The highest intensity band at 1525cm−1, medium intensity band at 1156 cm-1) and the band at 1005 cm-1 showed in control vte1 and tmt plants higher, while in vte4 mutants lower level, when compared to WT plants. Based on Raman spectra, in control condition, the amount of CARs in vte1 and tmt plants was higher and in vte4 mutants lower when compare to WT plants (Fig. 1A).

NaCl-treatment led to changes in Raman spectral pattern characteristic of CARs. All plants with altered TCs composition had lowered bands intensity, mainly at 1525 and 1156 cm-1 than WT plants (Fig. 1A). Interestingly, more stronger decline in these bands intensity showed vte1 mutants and tmt transgenic line than vte4 mutants when compare to parallel controls. Simultaneously, these bands (at 1525 and 1156 cm-1) was almost unchanged in WT plants. In contrast, the band at 1005 cm-1 of lowest intensity, showed a slight decline in tmt transgenic line, strong decline vte1 mutants, and stayed nearly unchanged in vte4 mutants and WT plants when compared to control plants. Under salt stress the decline in the amount of carotenoid compounds for tmt (~ 60%) and vte1 (over 40%) plants was stronger than for vte4 plants. In contrast, in WT plants, the level of CARs were comparable throughout the study under control and salt-stress conditions.

2.2. Chemometrics - Cluster analysis-Kindly elaborate it.

We have explained the concept of cluster analysis.

We added: “Cluster analysis is a method of classifying tested objects into groups (clusters) so that the resulting clusters contain objects that are as similar as possible.”

We also added:

“We used the cluster dendrograms to find meaningful and systematic differences among the measured spectra of the chemical compounds present in the plant tissues (Fig. 1B). The following dendrograms show the cluster analysis separately for two mutants (vte1, vte4), transgenic line (tmt) and WT plants grouping the control and salt-stressed plants (Fig. 1B). Distinct discrimination was achieved for the two groups: control and NaCl stressed plants for the entire measuring range for each genotype - with or without a-TC and WT. It was demonstrated that the leaves of vte1, vte4, tmt and WT plants differed significantly in their content and composition of carotenoids, chlorophylls or polysaccharides and that these differences depended on salt stress conditions (Fig. 1B).“

In order to clear up the reviewer's doubts, we would like to explain that the dendrograms, which were shown in the manuscript (Fig. 1B) are the usual way of representing the performed cluster analysis. This is how the authors presented the results of the cluster analysis in previous works on the effects of environmental stress on plants. As an example we can mention:

Rys M., Pociecha E., Oliwa J., Ostrowska A., Jurczyk B., Saja D., Janeczko A. 2020. Deacclimation of Winter Oilseed Rape-Insight into Physiological Changes. Agronomy 10: 1565.

Lukaszuk E., Rys M., Możdżeń K., Stawoska I., Skoczowski A., Ciereszko I. 2017. Photosynthesis and sucrose metabolism in leaves of Arabidopsis thaliana aos, ein4 and rcd1 mutants as affected by wounding. Acta Physiol Plant 39: 17.

Saja D., Rys M., Stawoska I., Skoczowski A. 2016. Metabolic response of cornflower (Centaurea cyanus L.) exposed to tribenuron-methyl: one of the active substances of sulfonylurea herbicides. Acta Physiologiae Plantarum 38: 168.

Synowiec A., Rys M., Bocianowski J., Wielgusz K., Byczyńska M., Heller K., Kalemba D. 2016. Phytotoxic effect of fiber hemp essential oil on germination of some weeds and crops . Journal of Essential Oil Bearing Plants 19: 262-276.

Rys M., Szaleniec M., Skoczowski A., Stawoska I., Janeczko A. 2015. FT-Raman spectroscopy as a tool in evaluation the response of plants to drought stress. Open Chemistry (Central European Journal of Chemistry), 13: 1091-1100.

The description of Fig. 1B was also rewritten as follows:

Cluster analysis of the FT-Raman spectra of the WT and TC-deficient mutants (vte1, vte4) and a-TC accumulated transgenic (tmt) line of Arabidopsis thaliana irrigated with water (control - c) or with NaCl solution (-s) after 10 days of the experiment.

Line 180-Statistically significant changes in 181 the total decline, as well as final level of Vx after de-epoxidation and total 182 Zx production, were observed between salt treated and control WT plants 183 (Fig. 3 A, B). It's not a clear message to me. Kindly write in a proper way.

We are very grateful for your valuable remark and we apologise for failed expressing our thought. We hope, it is more clear.

This part of the text has been rewritten.

Under salt stress, the initial rates of Vx de-epoxidation and Zx formation increased in WT, vte1 and tmt plants but decreased in vte4 mutants (Fig. 2A, B). Significant changes were determined for total Vx loss, calculated as a difference in Vx level at the beginning and end of de-epoxidation, and for the final level of Vx and total Zx production after de-epoxidation between salt treated and control WT plants (Fig. 3 A, B).

Line 187-The total decline and final level of Vx after de-epoxidation, were comparable with whom? among all group of plants subjected to salinity (Fig. 3A), while total Zx production was lower in transgenic tmt line in comparison with WT plants which one? Salt treated WT or Controlled WT? (Fig. 3B).

and

Line 192-De-epoxidation rate was so high, that maximal level of Ax accumulation increased at c.c. 2 pp and was the largest, 194 among of tested plants (Fig. 3C). You just said higher or increased but in which genotype?

We changed this paragraph as follows, hoping it's more readable:

The total decline and final level of Vx after de-epoxidation was similar in all tested experimental genotypes irrespective of the treatment (Fig. 3A). No significant changes were also observed in the total decline and final level of Vx after de-epoxidation between the mutants (vte1, vte4), the transgenic tmt line and WT plants following exposure to salt stress. Under salt stress the only significant differences were spotted between WT plants and tmt transgenic line regarding a decline in total Zx production after de-epoxidation (Fig. 3B). The highest initial rate of Ax formation among all tested plants was detected for salt-stressed vte1 mutants (Fig. 2C). Salt treated vte1 plants showed also by about 2 pp higher maximal Ax accumulation than control vte1 plants, whereas the differences between maximal Ax accumulation and the initial rate of Ax formation in the remaining variants were not significant, irrespective of NaCl presence. Salt-stressed vte1 mutants showed also the highest differences between maximal and final level of Ax during de-epoxidation (Fig. 3C).

2.7. Antioxidant enzymes activities

In figure 6B-The gel pictures are a blur and not able to see the clear band. Additionally, Controls are missing from this image. Therefore, need to replace it with a new picture along with negative and positive controls.

We improved the quality of the gel pictures showing SODs activities.

In our experiments all tested plants were divided into two sets: one set of plants was watered with tap water (controls), and the second one - salt stressed plants were watered with tap water but with an additional 200 mM of NaCl.

As in other experiments of this type, in our research all measurements done on Arabidopsis mutants and transgenic lines were compared with those on WT type (wild type), which play a role of control. To evaluate the effect of NaCl, on all studied genotypes (WT, mutants and transgenic line) as control we used measurements done on plants non-treated with NaCl. In the whole text in Results and Discussion manuscript we used these type of controls. In fact, chosen plants treated with salt can be compared with WT plants and chosen type of plant material, however to simplify interpretation of our results we concentrate on direct comparison of plant treated with NaCl versus non treated.

Line 263-Low activity of MnSOD and CuZnSOD isoforms was also 264 detected.???? You have just landed the sentences. Kindly write it properly.

and

Line 279-Very similar is its rank in respect of POX, but showing an increase in activity (WT 280 vte1 vte4 tmt). What?? I did not get. Kindly rewrite it.

We corrected these sentences as follows:

“The activity of Mn-SOD and CuZn-SOD was lower only in plants without a-TC.”

and

“NaCl treated plants can be ranked according to their CAT activity in the following order: WT>vte1=vte4=tmt, and according to their POX activity as: WT >vte1>vte4 >tmt.”

Discussion:

Line 284-Our studies also proved that A. thaliana plants are able to adapt to NaCl stress under light intensity How??

We have changed the statements as follows:

Our studies proved that the changed a- and g-TC content is not a critical prerequisite for the survival of A. thaliana plants growing under salt stress (200 mM NaCl) and light intensity not exceeding 120 μmol photons m-2s-1. A. thaliana plants with altered TC composition were capable of adapting to NaCl stress through modulation of their defense mechanisms that include not only TCs as quenchers of singlet oxygen formed at PSII.”

Line 289- Tocopherols composition modulate the level of photosynthetic pigments (chlorophylls, 290 carotenoids) and intensity of Vx-de-epoxidation

Where is the data of photosynthesis parameters of growing plants?

Thank you for your valuable comment.

We measured Fv/Fm in control and salt-stressed plants used in our experiments. The obtained data were presented in our pervious paper Surówka et al. (2020) which is cited in the manuscript. As these data were obtained on the same plant material, Fv/Fm parameter characterise also Arabidopsis plants used in this experiment.

We did not state the photoinhibition of PSII in salt-treated plants with altered TCs composition, and even observed a slight stimulation of Fv/Fm parameter (Fig. 1A).

Similar results, showing the lack of photoinhibition of PSII in A. thaliana plants with different TCs composition treated with NaCl were presented by e.g. Cela et al. (2011).

For the attention of the reviewer, we enclosed these results (Fig. 1A).

We added to the text of manuscript in “Introduction section” sentences:

Our previous study in Arabidopsis thaliana with altered TC composition exposed to salt stress showed a slight stimulation of the maximum operating efficiency accompanied by strongly altered cellular osmolarity (Surówka et al. 2020). Therefore, in this study we investigated the influence of TC composition on selected C-, N- and S-containing compounds with antioxidant and osmoprotectant properties that can be linked with TC based on their function or biosynthesis pathway (Diagram 1).

and in the “Discussion section” sentence:

“The changes in the level of photosynthetic pigments (chlorophylls, CARs) in A. thaliana plants with different TC composition subjected to salt stress (Fig. 1A) were accompanied by a slight stimulation of the maximum quantum yield of PSII (Surówka et al. 2020).

Line 291-Obtained Raman spectra revealed that mutual (a - and g-) TCs balance play role 292 in photosynthetically active pigments (chlorophylls, CARs) adjustment under both con293 trol and salt stress conditions (Fig. 1A, B). According to Ischebeck et al. [44] and Dörmann 294 [45]. This statement is not supported by your results.

We are very grateful for your remark. In addition to the improving the description of chlorophyll and carotenoid levels based on Raman spectra in Results section. We modified the sentences in discussion as follows:

The Raman spectra showing changes in the amount of chlorophylls and carotenoids that depended on TC composition in control and salt stressed plants (Fig. 1 A,B) revealed that mutual (a and g) TC balance is important for the adjustment of the photosynthetically active pigments under both control and salt stress conditions. The changes in the level of photosynthetic pigments (chlorophylls, CARs) in A. thaliana plants with different TC composition subjected to salt stress (Fig. 1A) were accompanied by a slight stimulation of the maximum quantum yield of PSII (Surówka et al. 2020). We therefore suppose that a dynamic reorganization/recycling of metabolites related to TC enabled the plants to survive salt stress. The interdependence of TCs and chlorophylls is probably related to chlorophyll and protein degradation as well as de novo synthesis of TC precursors. These processes contribute to the increased demand for substrates used in TC synthesis during abiotic stress (Collakova and DellaPenna 2003). As showed by Ischebeck et al. [44] and Dörmann [45], free phytol from chlorophyll breakdown under stress might be directly used for TC biosynthesis.

The results obtained on the basis of Raman spectra (carotenoids and chlorophyll content) are not directly related to the photosynthetic efficiency. Fluorescence‐based maximal quantum yield for PSII were presented and discussed in our previous work (Surówka et al. 2020). which is cited in this manuscript.

As you reported the role of these mutants in photosynthesis. What about the role of these in chloroplast morphology? Do you have electron microscopy pictures of chloroplast under control and salt stress?

Thank you for this valuable suggestion.

Tocopherols are functioning in plants as lipophilic antioxidants that protect cellular membranes form degradation. In plants, tocopherols are as lipophilic antioxidants, that are believed to protect chloroplast membranes from photooxidation and help to provide an optimal environment for the photosynthetic machinery (Munne-Bosch and Alegre 2002).

On the other hand, it has been shown in several papers (e.g. Collakova and DellaPenna 2003, Cella et al. 2011, Ellouzi et al. 2013, Asensi-Fabado et al. 2015, Surówka et al. 2020) and also in our work, that TCs affects levels of several metabolites, which at least, at the initial state take part in modulation of chloroplast metabolism and structure.

Previously, TCs-dependent changes in electron transport linked with alteration of thylakoids structure in chloroplasts in vte1 mutants exposed to high or low light were showed on electron micrographs by Niewiadomska et al. (2018).

We are currently preparing the project, in which we include the research task focused on the detailed analysis of changes in chloroplast structure/morphology. As chloroplasts cooperate with other organelles, including mitochondria and peroxisomes (what was also confirmed in this manuscript showing TCs-dependent changes in metabolites and antioxidative enzymes linked with these organelles), we are planning detailed analysis of alterations in structures/morphologies of these organelles.

Material methods:

Note-Kindly provides the transcript level of your transgenic lines under normal as well as under salt stress.

As our studies are generally focused on biochemical characteristic of Arabidopsis plants with changed TCs composition, the most metabolites, including tocopherols content were determined by HPLC. To characterise plant material used in our studies we added to M&M the information as follows:

The experimental plant line of A.thaliana used in our experiments has a well-documented origin (Desel et al. 2007) and was thoroughly characterized e.g. by Porfirova et al. (2002), Bergmuller et al. (2003), Shintani and DellaPenna (1998), Collakova and DellaPenna (2003), Rosso et al. (2003).

Porfirova, S., Bergmuller, E., Tropf, S., Lemke, R. and Dormann, P. (2002) Isolation of an Arabidopsis mutant lacking vitamin E and identification of a cyclase essential for all tocopherol biosynthesis. Proc. Natl Acad. Sci. USA 99: 12495–12500.

Bergmuller, E., Porfirova, S. and Dormann, P. (2003) Characterization of an Arabidopsis mutant deficient in gamma-tocopherol methyltransferase. Plant Mol. Biol. 52: 1181–1190.

Shintani D, DellaPenna D (1998) Elevating the vitamin E content of plants through metabolic engineering. Science 282:2098–2100.

Collakova E, DellaPenna D. The role of homogentisate phytyltransferase and other tocopherol pathway enzymes in the regulation of tocopherol synthesis during abiotic stress. Plant Physiol. 2003 Oct;133(2):930-40. doi: 10.1104/pp.103.026138. Epub 2003 Sep 25. PMID: 14512521; PMCID: PMC219066.

We also added sentences, and Table (1) showing a- and g-TCs content in A. thaliana plants used in experiment presented in this manuscript and also in our previous paper (Surówka et al. 2020) as below: The content of a- and g-TC in the rosette leaves collected at the end of the experiment in both control and salt stress conditions was detected according to a method of Surówka et al. (2016) and presented in Tab. 1.

A.thalianawith changed TC composition

/treatment

a-TC mg/g FW

Standard deviation (SD)

g-TC mg/g FW

Standard deviation (SD)

WT H2O

72.6

3.67

4.1

0.2

WT NaCl

83.5

6.05

6.5

0.8

vte1 H2O

ND*

ND

ND

ND

vte1 NaCl

ND

ND

ND

ND

vte4 H2O

ND

ND

58.0

3.7

vte4 NaCl

ND

ND

72.3

5.9

tmt H2O

115.5

7.51

ND

ND

tmtNaCl

134.7

11.68

ND

ND

Table 1. The content of a- and g-TC in the rosette leaves of A.thaliana with various TC composition growing in control and salt stress conditions.

In WT plants, the levels of α-TC were comparable throughout the study under control and salt-stress treatment (Tab. 1). As expected, vte4 mutant showed γ-TC but not α-TC accumulation in the rosette leaves, vte1 mutant did not accumulate either α- or γ-TCs in the leaves, and tmt transgenic line showed a-TC, but not g-TC accumulation in the rosette leaves in both control and salt stress conditions.

TCs content was measured based on the method reported by Surówka et al. (2016).

Surówka, E., Dziurka, M., Kocurek, M., Goraj, S., Rapacz, M. and Miszalski, Z. (2016) Effects of exogenously applied hydrogen peroxide on antioxidant and osmoprotectant profiles and the C3-CAM shift in the halophyte Mesembryanthemum crystallinum L. J. Plant Physiol., 200, 102–110.

As our results are in agreement with data presented by other authors, who showed the total TCs amount or the content of α- or γ-TCs in Arabidopsis plants exposed to high light (e.g. Collakova and DellaPenna 2003, Niewiadomska et al. 2018), or to salinity (e.g. Cella et al. 2011, Ellouzi et al. 2013), we did not characterise the transcriptome of studied plants. Such analysis is also planned in both control and salinity conditions as a part of the next research.

Reviewer 2 Report

The article has an interesting topic and novelty however the English of the article needs to be improved extensively because through grammatical and composition errors loses almost completely its scientific value. Therefore, I suggest the authors to check their article first with a native English speaker to make it worth reading to the end.

Author Response

Response to Reviewer 2

Dear Sir/Madam,

We are grateful for your suggestion of extensive improvement of the language of our manuscript. We apologise for sending the manuscript with numerous grammatical and stylistic errors.

The revised version of the manuscript was corrected by a professional Life Sciences editor and we hope that the text of the manuscript is now more comprehensible.

We are also grateful for drawing our attention to the importance of precise description of the results and facts in scientific papers, as well as the content of information in the abstract.

Following your suggestions, we rewrote the following sentences in the abstract:

“To determine the role of α- and γ-tocopherol (TC), the study compared the response to salt stress (200 mM NaCl) in wild type (WT) Arabidopsis thaliana (L.) Heynh., its two mutants: (1) totally TC-deficient vte1, and (2) vte4 accumulating γ-TC instead of α-TC, and (3) tmt transgenic line overaccumulating α-TC. Raman spectra revealed that salt-exposed α-TC accumulating plants were more flexible in regulating chlorophyll, carotenoid and polysaccharide levels than TC deficient mutants, while the plants overaccumulating γ-TC had the lowest levels of these biocompounds.”

We also modified the paragraphs focusing on the role of chloroplasts, to show more precisely the role of selected metabolic pathways in the synthesis of the metabolites discussed in our manuscript. Now, we hope, it is more readable and more informative.

“Chloroplasts, through the shikimate/phenylpropanoid pathway take part in the biosynthesis of chorismate, a precursor of aromatic amino acids (e.g. L-phenylalanine and L-tyrosine), and further intermediates such as homogentisic acid (HGA, polar precursor of tocopherols), as well as a number of phenolic compounds (Diagram 1). Through 2-Cmethyl-D-erythritol 4-phosphate (MEP) pathway, chloroplasts are the sites of biosynthesis of geranylgeranyl pyrophosphate (GGPP), an intermediate in the biosynthesis of (poly)isoprenoids, such as e.g. carotenoids (CARs), chlorophylls, plastoquinol-9 or lipophilic polyprenyl – a precursor of tocopherols (TC) [8-14]. All these metabolites are crucial for the molecular and physiological regulation of plant cell functioning, especially under stress conditions.”

Reviewer 3 Report

I would ask author to rewrite manuscript in order to be readable. Omit all unnecessarily long sentences full of abbreviations that are hard to read and follow. Also manuscript is written using different fonds, all what make me hard to follow. After these modifications I will be able to normally review the content of the paper.

Author Response

Response to Reviewer 3

Dear Sir/Madam,

We are grateful for your valuable comment regarding the correction of English in our manuscript. We thoroughly edited the manuscript. According to your suggestion, we especially focused on shortening long sentences and reducing the number of abbreviations used in the text. We apologize that the use of long sentences full of abbreviations made the text incomprehensible and we are sorry to present you with a manuscript written in different fonts.

The manuscript was corrected by a professional linguist. We unified the font used for the text of the manuscript.

We hope that the text of our manuscript is now clearer to read and more comprehensible.

Round 2

Reviewer 1 Report

The authors improved the manuscript as per the reviewer's suggestion. Hence, I would like to recommend this manuscript for publication.

Thank you

Author Response

Dear Sir/Madam,

Thank you for the revision of our manuscript, for the constructive comments that allowed the text to be more readable and understandable and for accepting our answers and corrections included in the manuscript.

Reviewer 3 Report

Manuscript is not submitted in the IJMS format and does not have numbered lines, so it is difficult to review. I am not sure why authors waste my time, first they put different funds in a manuscript and submitted manuscript with unreadable English now they submitted version which is not in MDPI format. Study and results are interesting but I do not know why authors does not want to prepare manuscripts according to Authors information pack. It will be easier for reviewers.

Figure 1A and Figure 1B should be Figure 1 and Figure 2. Those are to separate figures.

Did authors performed just one experiments or they repeat experiments several times? How they defined replicates?

It is weird that authors put diagram in conclusions section. This should be in discussion part.

Author Response

Dear Sir/Madam,

We would like to thank you for taking the time to review our publication. We are very grateful for your comments and suggestions, which helped us to improve the quality of the article. 
Changes introduced to the manuscript according to the Reviewer suggestions have been highlighted in green.
Our detailed responses to your comments are presented in the attached letter.

Round 3

Reviewer 3 Report

Now finally manuscript is in good format. In my opinion it could be published. Just before publication authors should change latin names of plants into italic. for example in abstract when first time mention full latin name Arabidopsis thaliana L. and in the rest of the manuscript A. thaliana. When they just use Arabidopsis there is no need for italic.